# A programmable chemical computer with memory and pattern recognition

Juan Manuel Parrilla-Gutierrez [1], Abhishek Sharma[1], Soichiro Tsuda[1], Geoffrey J.T. Cooper[1], Gerardo Aragon-Camarasa [1], Kevin Donkers[1] & Leroy Cronin[1 ✉]

Current computers are limited by the von Neumann bottleneck, which constrains the throughput between the processing unit and the memory. Chemical processes have the potential to scale beyond current computing architectures as the processing unit and memory reside in the same space, performing computations through chemical reactions, yet their lack of programmability limits them. Herein, we present a programmable chemical processor comprising of a 5 by 5 array of cells filled with a switchable oscillating chemical (Belousov–Zhabotinsky) reaction. Each cell can be individually addressed in the 'on' or 'off' state, yielding more than $2.9 \times 10^{17}$ chemical states which arise from the ability to detect distinct amplitudes of oscillations via image processing. By programming the array of interconnected BZ reactions we demonstrate chemically encoded and addressable memory, and we create a chemical Autoencoder for pattern recognition able to perform the equivalent of one million operations per second.

---

[1] School of Chemistry, University of Glasgow, University Avenue, Glasgow G12 8QQ, UK. ✉email: Lee.Cronin@glasgow.ac.uk

Over the last 50 years computers have become ubiquitous, essential for many aspects of modern life. During this period their processing power has increased manifold, but the paradigm used has remained the same keeping the processor and memory separate[1,2], creating a data bottleneck; and using binary state electronic switches[3], which are limited by the fabrication limits CMOS technology, and power dissipation[4]. Systems based upon quantum effects promise to solve problems intractable for conventional computers as the number of qubits increases[5,6], but they are yet to reach their full potential. However, nature exploits the parallelism of collective networks[7,8] by developing systems able to process information despite large amounts of noise[9]. Furthermore, it has been demonstrated that even non-living chemical systems can implement computations[10–15], yet their lack of programmability limits their practical use.

Most of the unconventional computational architectures are based on coupled-oscillator systems at different length scales. These include oscillator-based architectures that have been conceptualized and implemented to a certain extent utilizing chemical[16], chemomechanical[14], spin-torque[17,18], or laser-based oscillators[19]. Many oscillator-based computational architectures usually suffer from weak coupling, generating non-synchronous phases, and external noise effects. Substantial progress has been made to reduce these problems. Chemical computation architectures utilizing reaction-diffusion systems have demonstrated solutions to optimal paths for labyrinths[20,21], logic gates[22] and Boolean circuits[23]. In the particular case of the BZ reaction, computational architectures using it have been able to emulate logic gates[24] and complex circuits[25], perform image processing[26] and pattern recognition[27], solve optimization problems[28], control mechanical components[29] or soft matter[30], and create neuromorphic architectures[31]. All these problem-specific platforms utilize excitable chemical medium and observe time-evolution of spatiotemporal oscillations as a computational logic for information processing. Different possible architectures have been conceptualized in single-phase two-dimensional architectures where a problem is directly mapped[16] or in a multi-channel network[32] as well as in multiphase interconnected droplets or vesicles[33]. With the advent of high-performance 3D printing architectures and fast machine learning techniques for image processing, computation capabilities utilizing chemical systems can be significantly improved by introducing a hybrid approach. This includes using electronic actuators as inputs for problem instantiation and chemical evolution and digital image processing as an information processing unit.

Herein, we show a chemical processor that utilizes individually addressed, but fully interconnected cells of a chemical oscillating Belousov–Zhabotinsky (BZ) reaction as the data processing medium[34]. Furthermore, we show that the system can be programmed to achieve flexible and multi-purpose calculations by addressing the individually controlled stirrers and adjusting the stirring speeds (Fig. 1 and Supplementary Video 1). Our processor architecture relies on data storage and processing via electron transfer between molecules of $[Fe(Bpy)_3]^{2/3+}$ as a catalyst for the BZ reaction[35] and an indicator where the oxidized regions containing Fe(III) are blue, and the reduced states containing Fe(II) species are red. The output from the interconnected cells is produced by recording a video of the BZ medium to monitor the oscillation states of the reaction in the individual cells. To achieve programmability, we design a platform that controls the inputs as oscillations of the BZ reaction at local sites in a grid (cells) by externally controlling the oscillations in each cell with a magnetic stirrer, where cells are triggered when a stirrer is turned on or when an off cell is surrounded by on cells and the chemical oscillations transfer to it. The cells interact with each other via hydrodynamic coupling arising from the fluid flow created by magnetic stirrers. As a result, by controlling the input configuration of the magnetic stirrers, the platform can be programmed to exploit the chemical states arising from interactions between spatiotemporal excitation patterns. This is because the chemical oscillators behave like the coupled-oscillator model[36], where the excited waves generated at a cell can propagate to neighbouring cells by setting the local phase as well as the oscillation frequency of the active cells. The oscillation frequency can be controlled after an initiation time via the stirrer speed to form a globally synchronized oscillating pattern. Moreover, the stirring patterns can be individually changed at any time based on user input. Therefore, the BZ processor described here can be programmed at any point during its execution.

## Results

**BZ processing platform**. The automated BZ platform is designed so that it can be used as a programmable chemical processing system, exploiting the excitability and bistability of the oscillating chemical reaction. We demonstrate programmability by performing different processing tasks in our BZ platform, such as memory and pattern recognition. Our digital-chemical processing system consists of four components (see Fig. 1, Methods and Supplementary Methods 1). (i) The BZ reactor cell grid can be customized to the desired geometry depending on the experiment. In addition, by changing the size of the opening gap between neighbouring cells, it is possible to control the global propagation of the BZ excitation. When the gap size is large enough, the whole BZ medium generates coherent excitation wave patterns. (ii) A magnetic stirrer array consisting of 25 motors to control the stirrer bars within the BZ reactor cells. (iii) A control interface connected to a computer and the magnetic stirrer array. The rotation speed of each stirrer can be individually controlled; therefore, the local oscillations of the BZ reaction at a given cell can be individually addressed. (iv) The BZ reaction in the cells was monitored by a camera mounted above the grid, and the camera was connected to a computer enabling real-time analysis of the BZ. Each cell was classified as excited (blue) or non-excited state (red) as a function of time (see Supplementary Methods 2).

**Description of the BZ reaction**. Ferroin, $[Fe(Bpy)_3]^{2/3+}$, was chosen as the sole catalyst since this gives simple oscillations and the colour changes between the reduced form (red) and the oxidized form (blue) which are distinct and easily tracked optically. The other chemical components used were sulfuric acid, malonic acid and potassium bromate (see Methods and Supplementary Methods 3 for a description of how the solutions were prepared). To control the reaction, we exploit the fact that bulk oscillations of the BZ reaction break down and may become chaotic with short-lived or completely suppressed oscillations when they are not stirred[37,38]. Thus, the excitation of an individual BZ cell in the grid can be controlled by activating a stirrer placed in the cell (see Supplementary Methods 4).

To stir cells and drive patterns in the resulting oscillations, the platform described in Fig. 1 is used, where each cell contained a stirring bar and was directly placed above a motor with a pair of opposing magnets attached to its shaft. Using this mechanism, the stirring could be turned on and BZ oscillatory excitation waves generated only in specific cells. Defining which cells were stirred and which were not (i.e. input pattern) could then impact the formation of excitation wave patterns on the chemical system. This is because the stirred cells are more likely to generate oscillatory excited waves, which propagated to neighbouring cells and eventually formed a globally synchronized wave pattern. The speed of stirrers in the grid could be individually controlled and input patterns with different stirrer speeds could result in different global wave patterns of BZ reaction. Once a cell oscillates by being stirred, it will continue to oscillate long after the stirring is stopped

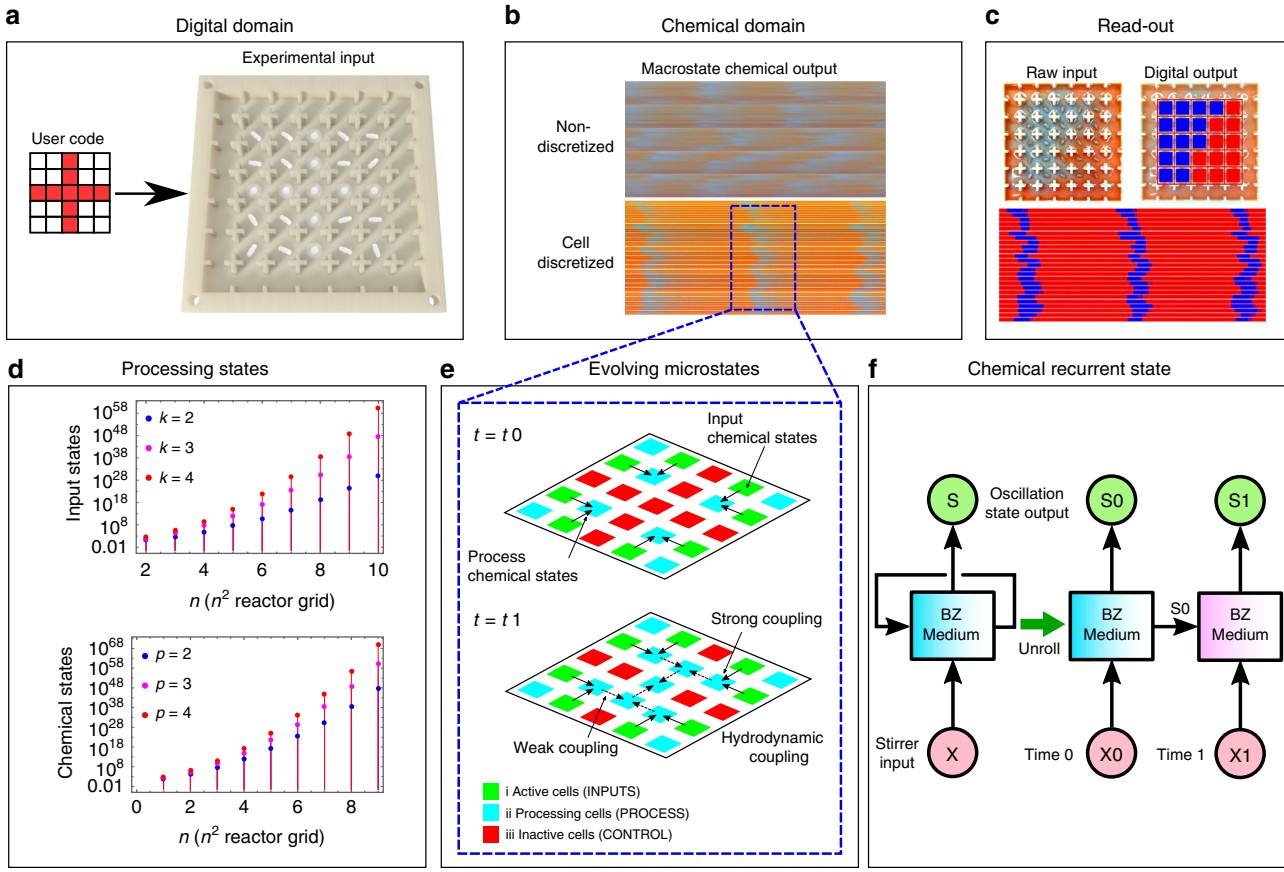

**Fig. 1 Chemical processor paradigm. a** Digital domain: The user inputs the actions to perform via a 5-by-5 matrix. This digital matrix is read by a computer and translated into machine code to mechanically actuate the stirrers at the user-defined speed. **b** Chemical domain: Based on the speed at which the stirrers rotate, the BZ reaction will oscillate between its two states, and oscillating waves will appear in the platform. If the arena does not contain physical boundaries between the cells, the oscillations will not localize and they would spread. If there are physical boundaries between the cells, the oscillations will localize at the specific cells within the arena. **c** Read-out: Using a camera and image processing, the states of the chemical processor are read by a digital computer. For every cell, the image processing algorithm will classify its state as on or off, depending on if the BZ reaction is oscillating on it. This step will return a 5-by-5 digital matrix representing the state of the system at each cell. **d** The Input States plot shows the scaling of the number of input states with the number of BZ cells on the experimental platform at different PWM stirring inputs ($k^{nxn}$, where $k$ can be 2, 3 and 4). The Chemical States plot shows scaling of the number of chemical states (defined by the distinct measurable amplitudes of BZ oscillations; see Supplementary Information) with the number of BZ cells on the experimental platform at a different number of measurable oscillation amplitudes ($(p+1)^{nxn}$ where $p$ describes the different detectable oscillation amplitudes and an additional one for no oscillation). **e** Evolving microstates: The cells in the arena are weakly connected; therefore, their oscillations will convolve, and be able to perform complex calculations by controlling the stirring speeds into (i) active cells—fast stirring—for inputs, (ii) process cells—slow stirring, (iii) inactive cells—no stirring (see Supplementary Information). **f** Chemical recurrent state: Because the BZ oscillations have memory, the global state of the medium not only depends on the input, but also on the state of previous iterations.

(see Supplementary Video 2). Moreover, the speed of each motor could be dynamically changed at any time, meaning that the platform can be programmable on demand.

Using grids of discrete but fluidically connected cells, we control the propagation of wave patterns from a cell to a neighbouring cell. The basic starting design for the platform comprised a $7 \times 7$ grid, where only the middle $5 \times 5$ cells were used. To control the interaction between cells, we first fabricated a prototype array of BZ cell grid using a 3D printer which had a v-shaped opening between cells. The BZ reaction volume used was 70 ml, which filled the arena to three quarters of its height, well above the v-shaped opening. With this design, it was found that oscillations did not propagate to neighbouring cells and the platform acted similarly to a display screen, where only the cells that were enabled flashed in blue, while the other ones remained red (Fig. 2a and Supplementary Video 3). To facilitate improved interaction between cells, we removed the v-shaped part of the opening, leaving only the corners of each cell to define it (Fig. 2b). This way the fluid from an activated cell would propagate to its neighbours

when stirred, and we could, for example, activate a cell that was disabled by stirring (and therefore activating) its neighbours.

**Pattern recognition using the BZ platform**. The wave propagation patterns described were found to be reproducible between parallel experiments, which indicates that the BZ platform can consistently convert an input pattern into a wave propagation pattern. Detection and interpretation of such propagating patterns were relatively easy with simple input patterns, but they increasingly become more complicated to human interpreters as the input patterns become complex. However, the complex patterns generated were neither random nor impossible to distinguish between them, because the BZ system generates consistent outputs in response to the same input pattern, and this is the basis for programmability. To prove programmability we were able to show that the global patterns being generated in the BZ reactor grid depend on the history of different patterns being inputted during execution as a function of time (see Fig. 3). Depending on the concentration of potassium bromate, the number of

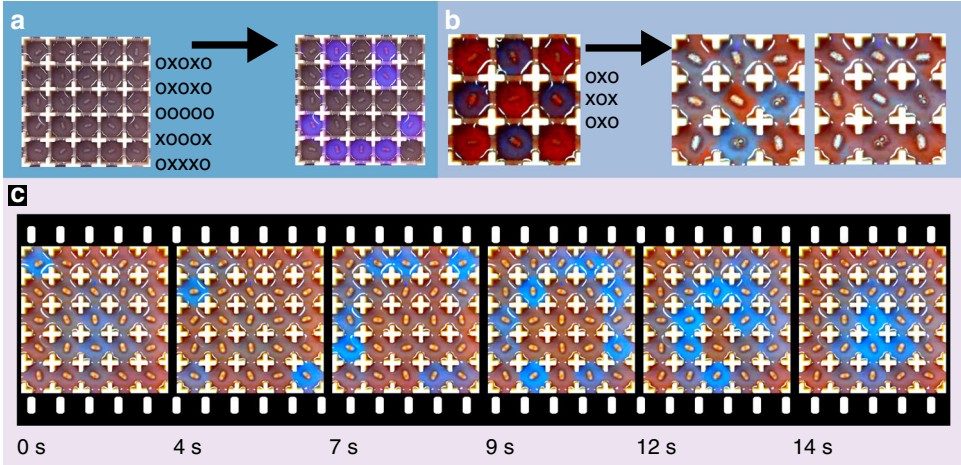

**Fig. 2 Controlling the oscillating cells.** In this figure, some frames are accompanied by a matrix of x's and o's. This matrix indicates if in that given frame a motor was enabled (marked with a x) or disabled (marked with a o). **a** With a v-shaped opening between cells, the oscillations generated at a given cell do not propagate to neighbouring cells. This way, the BZ reaction within each cell can be individually controlled, and the user can define which cells oscillate and which cells do not oscillate. **b** To increase the transfer of liquid between cells, the gap between cells was fully opened. This way we achieve the objective of enabling a cell which was disabled just by enabling its surrounding cells. **c** Once the cells are weakly connected, they will generate coherent patterns. (Note: the oscillations in the top-left image have been contrast-enhanced, Supplementary Movie 3 shows the unedited oscillations. See Supplementary Note 1 for more details.).

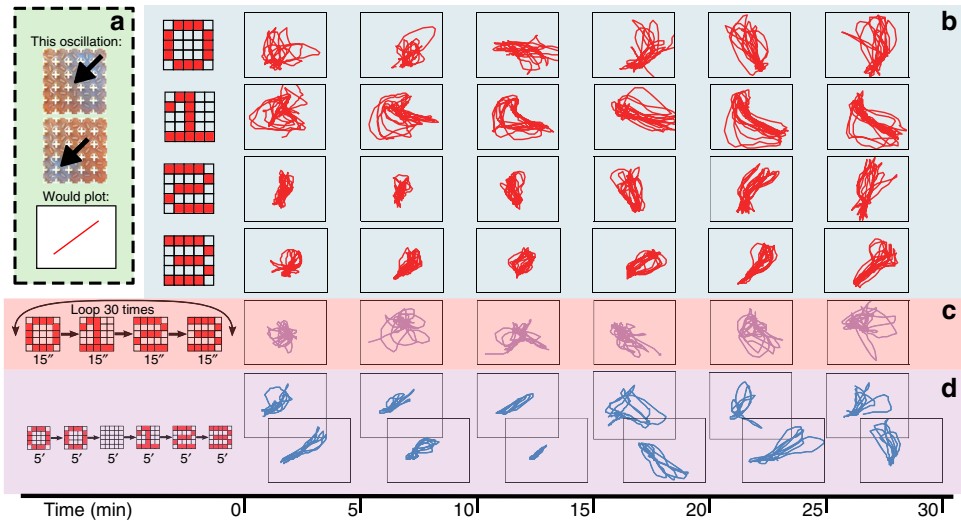

**Fig. 3 Programming the platform by introducing different input patterns during execution. a** For every frame in a video, the centre of mass (CoM) of a BZ oscillation was calculated using the blue channel of the frame, and then as these CoMs translated between frames, we plotted their translations using straight lines. Thus, each of the plots in this figure represents how the BZ oscillations moved across the arena. **b** In these four rows an experiment was performed where the same input pattern was applied to the BZ medium for 30 min. **c** In the following rows, two different experiments were performed where the pattern inputted to the BZ medium changed dynamically. In the first case, it iterated in a loop where every minute-long iteration inputted each of the four patterns in a sequence where each pattern was inputted for 15 s. **d** In the second case, each pattern was inputted only once: initially 0 for 10 min, then nothing for 5 min, and then 1, 2 and 3 for 5 min each.

oscillations that can occur without physical actuation can be up to eight repetitions (see Supplementary Information). This property can be used for short-term information storage, where the BZ platforms act similar to a volatile memory. By pairing the system programmability using the mechanical stirrers, and the system memory via the BZ medium, the platform keeps processing and memory residing in the same space.

To exploit the ability that the BZ system generates consistent outputs in response to the same input pattern, we adapt the 'reservoir computing' scheme[39] using the BZ platform (Fig. 4a). Namely, the system's output is interfaced with a neural network (NN), which is used to classify different input patterns based on the wave they generate. The generated wave propagation patterns

were input into an NN after being processed using image processing in order to identify the BZ oscillations (see Supplementary Methods 5 and Supplementary Video 4). Thus, this NN decodes the BZ outputs into human-readable data. To capture the temporal dynamics of the wave propagation patterns, a sliding window over 30 time points was used to create a dataset with 750 features (25 cells × 30 time points). We then use 20 different input patterns representing digital numbers, letters and random configurations (see Supplementary Methods 4), where the dataset was split to 30% between train and test set, and this gave correct answers with 80.5% accuracy against the test dataset. When using the same dataset but with 20 time points (thus 500 features), a split of 10% between the train and test set, and a deep

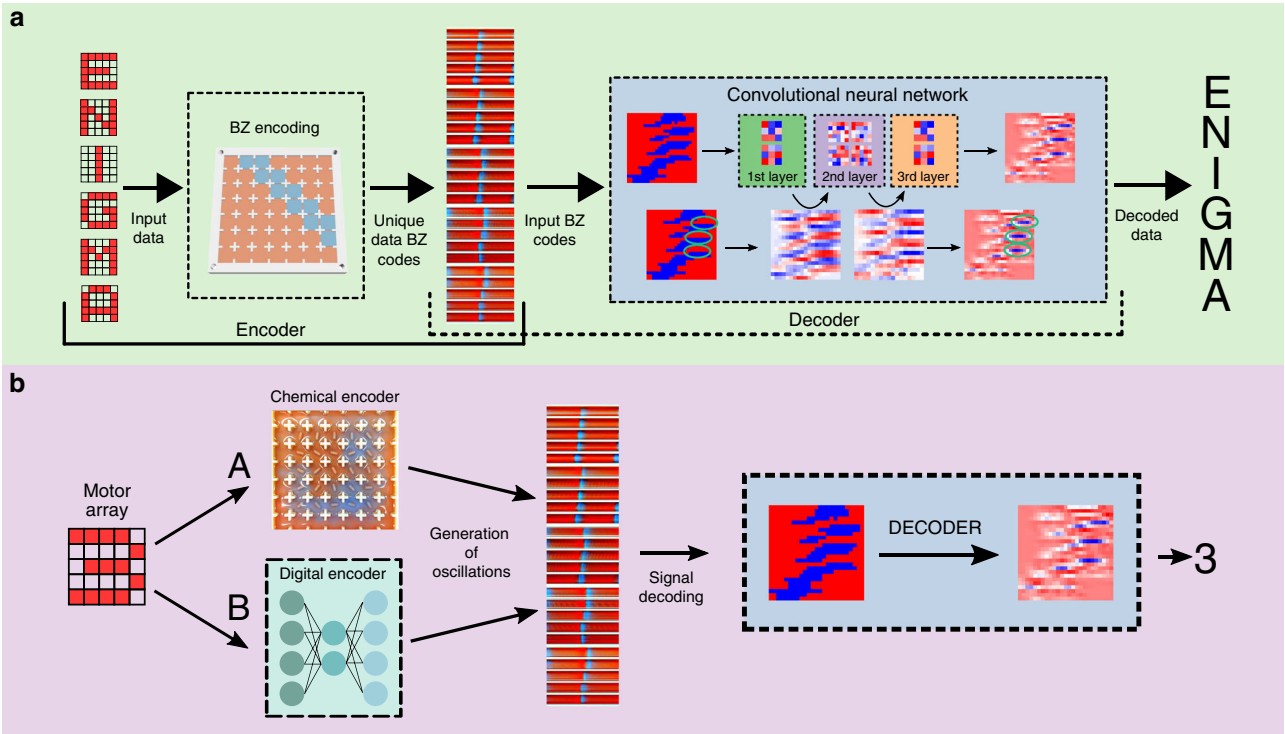

**Fig. 4 Pattern classification and data encoding by BZ reaction and machine learning. a** Schematic showing the full working pipeline of our system. Initially the user selects an input pattern, and this pattern is binarized in a 5-by-5 matrix. This matrix is used as a source for the PWM generator, and this will eventually generate a global oscillation. These oscillations are then decoded when needed for pattern recognition. **b** Flow diagram of the pattern recognition process comparing the BZ oscillations produced by a digital encoder and by the BZ platform. The A path uses the BZ medium as described above. The B path uses a digital encoder that has been trained beforehand in order to digitally produce BZ oscillations from motor patterns. Once the oscillations are generated using the BZ platform or the digital encoder, they are correctly decoded using a decoder as shown in (**a**). Both in (**a**) and (**b**), a snapshot of an oscillation is shown. The vertical axis of this snapshot represents each of the 25 cells, while the horizontal axis represents time. This snapshot was made juxtaposing consecutive frames to better represent the temporal domain of the BZ oscillations.

convolutional network (see Methods), the accuracy increases up to 92.5%. This result suggests that the BZ system produces consistent patterns upon a specific input pattern that can be distinguished by a machine learning algorithm (which may be too complicated for human eye). In other words, the BZ system satisfies the echo state property of reservoir computing[40]; i.e. each cell within its reservoir neighbours produces a nonlinear response signal which is combined into a desired output signal by means of linear combinations of these response signals in the BZ grid. In contrast, when using the reservoir system without the BZ reaction, but replaced with pure water, the output NN is not capable of classifying the patterns because the dynamics of the water are not sufficient to project input patterns into a higher dimensional space (see Supplementary Methods 4).

**Experimental benchmark of the chemical encoder**. To prove that during the pattern recognition process our BZ platform acts like a chemical processor, a similar experiment is performed, but in this case the BZ platform is replaced by a digital encoder that has been trained to digitally produce BZ oscillations from input motor patterns (Fig. 4b). The digital encoder uses the different motor patterns as inputs, and then digitally generates a BZ oscillation. Finally, this BZ oscillation is classified using the same NN that was just used as part of the 'reservoir computer'. The digital encoder is implemented following the architecture of an Autoencoder Neural Network (see Methods), and we estimate that, based on our architecture, it would require 60 million of logic gates for a forward pass through the network (see Supplementary Methods 6–8). Assuming that our BZ platform using 1 min of

window data encodes similarly to the digital encoder, then it decodes the equivalent of 1 million logic gates per oscillation.

This is effectively the same process detailed in Fig. 4a, but by-passing the BZ medium and using a digital encoder instead. The NN correctly labelled the different BZ oscillations, both the ones generated through the BZ medium, and the ones generated with the digital encoder. This unequivocally means that our platform is acting like a chemical processor because it is developing the same role as a digital encoder; specifically, we can consider this to be acting like a chemical encoder.

**Discussion**
In this work we have shown a programmable chemical processor driven by the oscillating BZ reaction and a reaction array of 25 switchable cells addressed by magnetic stirrers. The cells are fluidically connected so that the stirrer states affect both the local and global pattern. This means that information can be encoded into the chemical oscillators by the action of the stirrers, and then decoded by using video to record the states of the cells in the array as being in the oscillating on or non-oscillating off state. We showed how a near infinite number states can be used to both encode and decode letters and numbers, and that the chemical processor is able to recognize patterns with a low error rate.

The described chemical processor can serve as a sophisticated system in which the dynamics or functions that are theoretically difficult to infer can be calculated within the processing space defined by our BZ-driven chemical processor. For example, the system can be exploited as a chemical encoding device (see Fig. 4). In our tests, the system could distinguish a total of 20 patterns

reliably, with up to 92.5% accuracy. This was achieved by addressing the cells individually and generating oscillations at localized positions, yet they are weakly connected in a shared medium. The shared medium allows us to create a network of coupled oscillators, similar to, for example, the oscillatory neural computer network[41]. Some of the main limitations of the platform when compared to a digital computer are related to the chemistry itself. Firstly, the reagents that drive the BZ deplete as a function of time. In the experiments described here, once the reaction was prepared, it was left to stabilize during 10 min because in this window the oscillations were not visible, and then we focused on the following 30 min window where the oscillations were easily visible. Finally, the computations happen at the oscillation rate, which in our case was between 40 s and 1 min, which is slower than a digital computer. Nevertheless, the frequency of oscillations can easily be modified by changing the ratio of the different BZ components, and the number of operations per oscillation is potentially much higher considering the high parallelism of chemistry.

In the research described here, an automated platform containing magnetic stirrers is used to input information in the chemical processing medium, while a camera is used to read-out. This read-out step is assisted by an artificial neural network, where some of the processing is distributed in order to untwine the chemical states. The role of electronics could be further reduced by miniaturizing the BZ cells size so that cell—cell interactions occur spontaneously via reaction-diffusion travelling waves instead of hydrodynamic coupling using stirrers. The additional information processing using artificial neural networks could be reduced by evolving the experimental architecture (cell-to-cell interactions) in a computer using machine learning algorithms for pattern recognition. We propose that these dependencies could be removed stepwise, to move from a hybrid chemical computer towards a full autonomous one, where only the inputs and outputs are electronic, avoiding the all-or-nothing that currently plagues the study of unconventional computing. We believe that the platform and experimental work presented could result in a viable new perspective in the study of different computational paradigms beyond the von Neumann architecture and silicon-based computation. Not only does this open the prospect of developing chemical computers, the system described here should lead to new algorithm designs that could uniquely exploit the features of a chemical computer that uses electron transfer to compute over a range of length scales. Such systems have a vast number of states, are low power, but finding ways to design computing beyond the current silicon-based architectures will be required.

## Methods

**Automated platform.** An automated platform was designed and built in order to perform the experiments. The CAD design was done using OpenSCAD, and the pieces were manufactured using the 3D-Printer 'Stratasys Connex' and their material 'VeroWhitePlus'. Each cell in the platform contained a 8-by-3 mm stir bar. Under each cell there was a geared DC motor (6 V 200RPM). These DC motors were controlled with a PWM signal. This PWM signal was generated using Arduino Mega, and the shield from Adafruit '16-Channel 12-bit PWM/Servo shield —I2C interface'. A computer connected to the Arduino board through USB defined the different PWM signals through a Python script.

**Preparation of solutions.** Ferroin indicator: A stock solution of 1.0 M was prepared by dissolving 5.406 g of 1,10-phenanthroline in 10 mL water and adding 2.60 g ferrous sulfate hexahydrate while stirring. This stock solution was diluted to 0.1 M. Finally, 15 mL of this product was diluted in 95 mL of water for a total volume of 110 mL. Potassium bromate: 0.5 M solution was prepared by dissolving 8.35 g $KBrO_3$ in 100 mL of 1 M $H_2SO_4$. Sulfuric acid: 1 M $H_2SO_4$ was prepared by taking 5.6 mL conc. $H_2SO_4$ (96%, 18 M) and adding to water to reach a total volume of 100 mL. Malonic acid: 1 M solution was prepared by dissolving 10.4 g malonic acid in 100 mL of water.

1 M $H_2SO_4$ was sourced from Fisher Scientific, >95% analytical reagent grad. Malonic acid was sourced from Sigma-Aldrich, Reagent Plus 99%. Ferrous sulfate heptahydrate was sourced from Sigma-Aldrich, ≥98%. 1,10-phenanthroline was sourced from Sigma-Aldrich, ≥99%. Potassium bromate was sourced from different sources. The main body of work of this research used the one sourced from

Lancaster, 99%. Other suppliers used: Scientific Laboratory Supplies, 99% Alfa Aesar, 99.8% Alfa Aesar and Millipore Ensure.

**Image processing of recorded BZ reaction data.** The experiments were recorded using a web camera (LifeCam Cinema, Microsoft) and saved as AVI videos using MP4 compression, 800-by-600 pixels and 30 FPS. To classify the cells between red and blue, a dataset was built where a human labelled different cells as red or blue. An SVM was trained using this dataset. Then, each frame of a recorded video was processed by a pre-trained Support Vector Machine model to determine the activation/non-activation state of BZ cells (Fig. 1). The SVM library used was the one built into OpenCV. The activation state data categorized by SVM was exported into a CSV file, in which the activated state was defined as '1' and the non-activated state as '0'.

**Data preparation for pattern recognition.** Each 30′ experiment was binarized using an SVM as described. These 30′ videos were sped-up by a factor of 15 using Blender to obtain 2-min videos. Considering that the videos were recorded at 30 FPS, this generated videos with 3600 frames. A CSV was then built where every column represented a frame, and each row represented one of the 25 cells. Each position in this CSV was '0' if the cell did not oscillate at that time, or '1' if it did oscillate. This process was repeated for 20 different input patterns (see Supplementary Information), which were repeated for each pattern three times, generating a total of 60 CSV files. For each CSV file the initial 1000 columns were discarded. Then, using a moving window of size 30 (frames) by 25 (cells) sized windows (thus 750 features) were extracted and were kept if at least 100 of their points were set to 1. Following this procedure a total of 2236 data entries were produced.

**Pattern recognition using a convolutional neural network.** To build the convolutional neural network (CNN), Tensorflow 1.15 was used. The data entries just described were split 30% between the train and the test set (1565 for the training set and 671 for the test set). These data entries were the ones used to train the CNN, with an accuracy of 80.5% over the test set. This CNN contained three convolutional layers with 32 filters of size 5, 64 of size 3, and 128 of size 3. Finally, it had a fully connected layer with 64 neurons. In order to increase the accuracy to 92.5%, a train/test split of 10% was used, using a moving window size of size 20 (frames) by 25 cells (thus 500 features). The architecture of this network was as follows: Conv1 16 feature maps, kernel of size 5, dropout 50%. Conv2 identical but no dropout. Conv3 identical with dropout 50%. Maxpool layer with kernel of size 2 by 2 and strides of size 2 by 2. Conv4 32 feature maps and kernel of size 5, dropout 50%. Conv5 identical but no dropout. Conv6 identical with dropout 50%. Maxpool layer with kernel of size 2 by 2 and strides of size 2 by 2. Conv7 64 feature maps and kernel of size 3, dropout 50%. Fully connected layer with 64 neurons. Softmax layer.

**Convolutional neural network and Autoencoder.** To build the CNN and the Autoencoder, Tensorflow 1.15 was used. Patterns representing numbers from 0 to 9 were used. The inputs used were the CSV as described. These CSV contained 3600 columns values that represented 30 min of experiment. The first 1000 rows were discarded. The window size used was 25. Therefore, the inputs to the CNN were 25 by 25 or 625 values. The test and train ratios were set to 30%. Three CNN layers were used. The first one uses two feature maps and a kernel size of 3. The second one uses two feature maps and a kernel size of 5. The third one uses one feature map and a kernel size of 3. Finally, there is a fully connected layer with 16 neurons, and then 10 output neurons. The Autoencoder used had 25 inputs and 625 outputs. Three hidden dense layers were used of 50, 10 and 50 neurons. The Autoencoder was trained using two cycles of 3000 iterations each. The learning rate was 0.000001 and L2 regularization was used (0.00001). The input of this Autoencoder was 5-by-5 motor patterns, and its output of 625 nodes was used as input to the CNN described.

## Data availability

The data that support the findings of this study contain multiple video files for a total size of 1 TB. We have curated the dataset and it is available from the corresponding author upon reasonable request.

## Code availability

The source code is available at https://github.com/croningp/BZ1.

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

## Acknowledgements

We would like to thank Dario Caramelli for his help in producing some of the supplementary movies, and Andrew Quinn for some of the experimental work. The authors gratefully acknowledge financial support from the EPSRC (Grant Nos. EP/H024107/1, EP/I033459/1, EP/J00135X/1, EP/J015156/1, EP/K021966/1, EP/K023004/1, EP/K038885/1, EP/L015668/1, EP/L023652/1), the ERC (project 670467 SMART-POM), and the DARPA molecular informatics project.

## Author contributions

L.C. conceived the original idea and J.M.P.G., S.T., A.S. and L.C. together designed the project and the research plan; L.C. designed the reactor array and J.M.P.G., G.J.T.C. and K.D. designed and built the robotic platform, J.M.P.G. and G.A.-C. implemented the computer vision and the algorithms. J.M.P.G. performed the experiments. S.T., A.S. and J.M.P.G. did the data analysis. A.S. helped benchmark the system with L.C. and calculate the number of input and chemical states. J.M.P.G. and L.C wrote the paper with help from the rest of the authors.

## Competing interests

L.C. is listed as an inventor on a patent application filed by The University of Glasgow (GB 1815424.5).
