## [Peer Review File · Nature Communications]

REVIEWERS' COMMENTS:

Reviewer #1 (Remarks to the Author):

The paper "A programmable chemical computer with memory and pattern recognition" by Gutierrez et al formulates the new constructive paradigm for computer design. This paradigm is based on the oscillating chemical reactions, and the chemical medium is organized by well-known Belousov-Zhabotinski (BZ) reactions.

The idea of 'chemical computer' was formulated more than 30 years ago, however only two years ago the BZ-system was proposed as the basis of 'chemical computation' (see Gizynski, K. & Gorecki, J. Chemical memory with states coded in light controlled oscillations of interacting Belousov-Zhabotinsky droplets. *Phys. Chem. Chem. Phys.* 19, 6519–6531 (2017). It was referred in the Gutierrez et al paper)

Certainly, the constructed 'chemical computer' with one million operations per second (megaflops) is the slow one in comparison with the modern supercomputers which characteristics are in the domain between gigaflops and teraflops. However, the new technology is still at the starting point. Potentially, the 'chemical computer' will be characterized by more than 10^{17} chemical states. Therefore, advantages regarding the computer memory are obvious.

The paper presents different stages of computer construction, including

1. Design of the platform for controlling the inputs as BZ-oscillations at local sites
2. Pattern recognition. It was shown that wave propagation patterns were reproducible between parallel experiments.
3. Proof of the primary idea that the pattern recognition via BZ- platform acted like a chemical processor. This proof was done via replacing the BZ-platform by a digital recorder.

Generally, I've found that authors' reasoning is quite convincing.

I consider that the reviewed paper is the real step towards to the realization of the challenging alternative idea of 'chemical computer'.

Summing up, I recommend this paper for the publication in "Nature Communications".

Prof. Gregory S Yablonsky

Reviewer #2 (Remarks to the Author):

This is very well written paper which presents advances in the experimental implementation of chemical computing with BZ reaction. Citations to relevant works are very sparse -- plenty of papers have been published. With regards to the implementation, too much electronic devices are involved in the design of this BZ computing device -- this basically annihilate all potential benefits of the chemical computer. I urge authors to address this deficiency somehow. Another issue: BZ compartments are just playing roles of couple oscillators (any other material oscillators will do the job equally well).

Reviewer 1 Comments:

The paper "A programmable chemical computer with memory and pattern recognition" by Gutierrez et al formulates the new constructive paradigm for computer design. This paradigm is based on the oscillating chemical reactions, and the chemical medium is organized by well-known Belousov-Zhabotinski (BZ) reactions.

The idea of 'chemical computer' was formulated more than 30 years ago, however only two years ago the BZ-system was proposed as the basis of 'chemical computation' (see Gizynski, K. & Gorecki, J. Chemical memory with states coded in light controlled oscillations of interacting Belousov-Zhabotinsky droplets. Phys. Chem. Chem. Phys. 19, 6519–6531 (2017). It was referred in the Gutierrez et al paper)

Certainly, the constructed 'chemical computer' with one million operations per second (megaflops)) is the slow one in comparison with the modern supercomputers which characteristics are in the domain between gigaflops and teraflops. However, the new technology is still at the starting point. Potentially, the 'chemical computer' will be characterized by more than 10^{17} chemical states. Therefore, advantages regarding the computer memory are obvious.

The paper presents different stages of computer construction, including

1. Design of the platform for controlling the inputs as BZ-oscillations at local sites
2. Pattern recognition. It was shown that wave propagation patterns were reproducible between parallel experiments.
3. Proof of the primary idea that the pattern recognition via BZ- platform acted like a chemical processor. This proof was done via replacing the BZ-platform by a digital recorder.

Generally, I've found that authors' reasoning is quite convincing.

I consider that the reviewed paper is the real step towards to the realization of the challenging alternative idea of 'chemical computer'.

Summing up, I recommend this paper for the publication in "Nature Communications".

Prof. Gregory S Yablonsky

We appreciate the comments from this reviewer and his recommendation for the paper to be published in Nature Communications.

Reviewer 2 Comments:

This is very well written paper which presents advances in the experimental implementation of chemical computing with BZ reaction. Citations to relevant works are very sparse -- plenty of papers have been published.

As requested by the reviewer, the introduction of the manuscript has been extended with additional references and details of chemical computation architectures specifically to BZ

reaction. In particular, we have added 6 new references, and this section is highlighted o yellow in the introduction.

With regards to the implementation, too much electronic devices are involved in the design of this BZ computing device -- this basically annihilates all potential benefits of the chemical computer. I urge authors to address this deficiency somehow.

The authors like the mention that the Chemical Computing architecture proposed and designed in the current manuscript is a hybrid architecture. Electronic control provides precise control of operations in the chemical medium such as problem instantiation and readout. Here, problem instantiation (such as pattern recognition) occurs from the electronic medium into the chemical system where information processing occurs and later the oscillatory output is read by the imaging system and interpreted as the solution.

The current architecture is very simple and demonstrates the capability of a chemical system to perform useful computation. The role of electronics could be further reduced by miniaturizing the BZ cells size so that cell-cell interactions occur spontaneously via reaction-diffusion travelling waves instead of hydrodynamic coupling using stirrers. The additional information processing using Artificial Neural Networks could be reduced by evolving the experimental architecture (cell-to-cell interactions) *in-silico* based using machine learning algorithms for pattern recognition. However, the authors still believe that problem instantiation and readout will still be electronic.

Nevertheless, this was already addressed in the “Discussion” section of the paper, and we extended it with a bit more of explanation.

Another issue: BZ compartments are just playing roles of couple oscillators (any other material oscillators will do the job equally well).

It is correct that BZ compartments are playing roles of coupled oscillators and materials showing similar properties, in principle can perform the job equally well. There are various other systems equally capable such as spin-torque oscillators, relaxation oscillators based on phase-change materials. Spin-torque oscillators require much complex architecture (multilayer structure with metallic layers of Ta, Cu, Co, Ni, Au etc.) created using modern nanoscale fabrication techniques. Similarly, metal to insulator phase transition oscillators requires much complex electronic circuits and materials such as VO₂. Spin-torque oscillators usually have short-range spin-wave coupling and are more susceptible to noise and other environmental variations. However, our architecture is much simpler as compared to other oscillator-based computing architectures, scalable and can be easily miniaturized.